# A Low-Power High-Data-Transmission Multi-Lead ECG Acquisition Sensor System

**DOI:** 10.3390/s19224996

**Published:** 2019-11-16

**Authors:** Liang-Hung Wang, Wei Zhang, Ming-Hui Guan, Su-Ya Jiang, Ming-Hui Fan, Patricia Angela R. Abu, Chiung-An Chen, Shih-Lun Chen

**Affiliations:** 1Department of Microelectronics, College of Physics and Information Engineering, Fuzhou University, Fuzhou City 350108, China; N171127109@fzu.edu.cn (W.Z.); guanminghui16@163.com (M.-H.G.); jiangsy@landicorp.com (S.-Y.J.); 2Department of Information Systems and Computer Science, Ateneo de Manila University, Quezon City 1108, Philippines; pabu@ateneo.edu; 3Department of Electrical Engineering, Ming Chi University of Technology, New Taipei City 24301, Taiwan; joannechen@mail.mcut.edu.tw; 4Department of Electronic Engineering, Chung Yuan Christian University, Taoyuan City 32023, Taiwan

**Keywords:** multi-lead, wearable electrocardiogram (ECG) sensor system, Bluetooth, Huffman coding, low power consumption

## Abstract

This study presents a low-power multi-lead wearable electrocardiogram (ECG) signal sensor system design that can simultaneously acquire the electrocardiograms from three leads, I, II, and V1. The sensor system includes two parts, an ECG test clothing with five electrode patches and an acquisition device. Compared with the traditional 12-lead wired ECG detection instrument, which limits patient mobility and needs medical staff assistance to acquire the ECG signal, the proposed vest-type ECG acquisition system is very comfortable and easy to use by patients themselves anytime and anywhere, especially for the elderly. The proposed study incorporates three methods to reduce the power consumption of the system by optimizing the micro control unit (MCU) working mode, adjusting the radio frequency (RF) parameters, and compressing the transmitted data. In addition, Huffman lossless coding is used to compress the transmitted data in order to increase the sampling rate of the acquisition system. It makes the whole system operate continuously for a long period of time and acquire abundant ECG information, which is helpful for clinical diagnosis. Finally, a series of tests were performed on the designed wearable ECG device. The results have demonstrated that the multi-lead wearable ECG device can collect, process, and transmit ECG data through Bluetooth technology. The ECG waveforms collected by the device are clear, complete, and can be displayed in real-time on a mobile phone. The sampling rate of the proposed wearable sensor system is 250 Hz per lead, which is dependent on the lossless compression scheme. The device achieves a compression ratio of 2.31. By implementing a low power design on the device, the resulting overall operational current of the device is reduced by 37.6% to 9.87 mA under a supply voltage of 2.1 V. The proposed vest-type multi-lead ECG acquisition device can be easily employed by medical staff for clinical diagnosis and is a suitable wearable device in monitoring and nursing the off-ward patients.

## 1. Introduction

With urbanization and an accelerating population of the aging society, combined with a human unhealthy diet, lack of exercise, stress, and other factors, the number of people suffering from cardiovascular diseases continues to increase [1]. An electrocardiogram (ECG) is a common tool for evaluating whether a heart is healthy or not, which needs at least three leads of data for clinical diagnosis [2,3]. Since its introduction in the 1960s by the Massachusetts Institute of Technology Media Lab, wearable technology has received much attention and has become a hot topic of research [4].

In recent years, the development of microelectronics technology, Internet of Things (IoT) technology, and communication technology in medical applications have provided a good foundation for the development of wearable ECG equipment [5]. Several ECG monitoring devices have been proposed to monitor heart diseases [6,7,8,9,10]. Wearable devices are mainly used for two aspects. One is for fitness and exercise, where it is used to monitor the heart rate, steps, and other motion parameters to arrange reasonable movement for users. The other aspect is in the medical field, where it is used for medical detection and diagnostic purposes [11].

Wearable medical devices typically combine data acquisition devices with every day use. K. Takahashi et al. [12] designed a novel wearable device that uses a dry electrode on the elastic cuff of the garment to monitor the ECG of children. The University of Manchester developed a low-power wristband to monitor the ECG between the hands [13]. S. Majumder et al. [14] also proposed a wearable wristband ECG monitoring system that transmits the ECG data to a personal computer over the low-power Bluetooth (BT) communication medium. V.P. Rachim et al. [15] proposed a healthcare monitoring system implemented in an armband. These are single-lead wearable devices that are not recognized clinically for heart disease. The device should have at least three leads to acquire acceptable ECG information for disease diagnosis. The jacket developed by the University of Aveiro in Portugal is a wearable vital sign-monitoring system, which monitors a multi-lead ECG [16]. W.V. Rosenberg et al. [17] have developed a smart helmet that is especially useful for recording and monitoring vital signs in uncertain or dangerous situations, such as in racing or military activities. However, the power consumption of these two devices is relatively high. Low power consumption means that the device can work and operate for a longer period of time, which is particularly important for long-term monitoring of the physical condition of the wearable device [18]. These devices monitor human physiological signals, such as respiratory rate, heart rate, electrocardiograms, skin temperature, physical movement, brain activity, and blood pressure. Moreover, these devices transmit these physiological signals wirelessly to devices or networks for health monitoring, disease diagnosis, or treatment purposes [19].

At present, the use of wearable devices are on the rise. Power consumption is a main consideration for biomedical systems. Chen et al. [20] minimized the power by incorporating a micro control unit (MCU) to manage the power for each component in a biomedical system. Additionally, devices with multi-sensors are being promoted more than ever. A MCU with power management and a filter component in the reconfigurable circuit were provided in [21] to adjust signals in real-time. Moreover, Chen et al. [22] proposed systems for ECG data compression, particularly implementing the main aim of dropping power loss. In wireless systems, dropping the data transmitted by using fuzzy-control and entropy coding in stages reduced the data significantly. The resolution method, in which the fuzzy method is implanted, can improve the data compression performance efficiently. Chen et al. [23] not only improved the analog circuit, but also the analog-to-digital converter (ADC) by adaptively adjusting its sampling rate. This design will benefit both the limiting of data from the transmission and the minimizing of power of the analog and digital components.

This work has designed a low-power multi-lead wearable ECG signal sensor system that can simultaneously acquire the electrocardiograms of three leads and calculate the ECG of the seven leads of I, II, and V1, and the other four leads, III, aVL, aVF, and aVR, using Einthoven’s law [24]. Compared with the single-lead wearable ECG device, the designed device in this work uses more leads with a sampling rate of 250 Hz per lead, which has a higher clinical value. In order to meet the low-power requirements of wearable devices, in terms of hardware design, several low-power chips were incorporated. In terms of software, the working mode of a MCU was optimized to reduce the power consumption; a Bluetooth 4.0 low energy (BLE) protocol stack was modified to improve the parameters in the Bluetooth digital communication process; ECG data was compressed by Huffman coding to reduce the amount of data transmitted by Bluetooth and reduce the power consumption in a wireless communication process.

The rest of this paper is organized as follows. Section 2 presents the equipment architecture of the ECG monitoring sensor system. The details of the ECG vest and its low power design are described in Section 3. Section 4 presents the experimental results of the compression algorithm verification and low-power test. Section 5 concludes this paper.

## 2. System Architecture

The structure diagram of the multi-lead wearable ECG sensor system designed in this paper is illustrated in Figure 1, which includes an ECG vest and an ECG signal acquisition device. The core equipment consists of four components as follows: The power supply module, the acquisition module, the MCU module, and the Bluetooth communication module. The entire device was designed to meet the requirements of a portable ECG device for wearable devices with low power consumption and high accuracy measures.

The ECG test equipment can be worn on the human body using the arm band [15], the test clothes, or the chest strap [11]. Considering the comfort level of the user, the ECG test vest was selected as the front-end interface. The ECG test clothing needs to have a certain elasticity to ensure that the electrodes are closely attached to the human body. The vest uses five electrode patches to acquire the analog ECG signal from the surface of the human body. The ECG signal obtained is sent to the ECG signal acquisition module of the core device.

The overview of the ECG vest is shown in Figure 2. The front of the vest is designed with a zipper layer and a pocket to place the core device. There are five electrode patches on the vest, corresponding to the five positions of the right arm (RA), left arm (LA), left leg (LL), right leg (RL), and right chest V1 on the human body. The vest uses the RA and LA, RL and LL, and a Wilson center terminal (WCT) and V1 to acquire the analog ECG leads, I, II, and V1, respectively. The WCT is generated by the three patches of RA, RL, and LL. The RL patch sends a signal to the right leg drive (RLD) node of the ECG device, which is intended for control of the common-mode level of the patient. It is connected through the electrodes to the ADC chips, and thereby improves the alternating current (AC) common mode rejection ratio (CMRR) of the overall ECG system.

Compared to the traditional AgCl wet electrode patch used in most ECG instruments, the ECG vest designed in this work uses a silicone dry electrode that provides a comfortable feeling when attached on the skin of the user. Increasing the comfort level would be more satisfying if it met the requirement of a long-term monitoring of wearable devices. The silica dry electrode design is illustrated in Figure 3. The silicone dry electrode is integrated in the interior of the vest, and the bottom surface is in direct contact with the human skin ECG signal acquisition; the top surface of the dry electrode is in contact with the conductive button. In this way, the ECG vest can collect the ECG signal of the three leads, I, II, and V1.

The overview of the ECG signal acquisition device with a 250 Hz-high sample rate per lead is shown in Figure 4. The area of the printed circuit board is 7.2 cm × 6.3 cm. It consists of four parts; the power supply, the front-end circuits of the ECG signal acquisition and amplification, the MCU, and the Bluetooth 4.0 communication module [25].

The entire device is powered by a 500-mA lithium battery and can be charged through the micro universal serial bus (Micro-USB) interface. Figure 5 presents the connection diagram of the whole device.

The multi-lead wearable ECG device obtains the ECG signals of the three leads, I, II, and V1, from the surface of the human body through the ECG vest. The front-end chip is used to amplify, sample, and convert the ECG signal acquired from the body from an analog to a digital signal. The transmission of the ECG data to the MCU is done through a serial peripheral interface (SPI). Similar to the MCU, it controls the front-end ADC chip register configuration, reads the ECG data from it, and performs Huffman compression processing in its central processing unit (CPU) for ECG data size reduction. This allows the sampling rate to be improved to achieve higher data transmission and better wave performance.

The multi-lead wearable ECG signal sensor system presented in this work is designed to be portable. Recently, wireless ECG monitoring systems have been implemented [26,27] using Bluetooth and ZigBee as their main wireless communication protocols. WiFi communication protocols are widely used due to their advantages of having a wide range of coverage, fast transmission speed, and high security for a wide range of application. However, WiFi transmission requires large power consumption, which is costly and requires a router or an access point (AP) device for transmission. Hence, it is difficult to use WiFi communication protocols in wearable ECG devices. ZigBee, on the other hand, is an extension of the standard IEEE802.15.4 wireless personal area network (WPAN) protocol. It is mainly used in the field of automatic control and remote control [28]. It can be embedded in various devices and has the advantages of having low power consumption and low cost and has a strong networking capability. However, it has a low transmission rate. With the increase in the number of ECG leads and sampling rates of a device, ZigBee is not suitable as a communication protocol to be used to transmit the large amount of data due to its low communication rate. Although Bluetooth has a slow transmission speed, it is much faster than ZigBee. The speed of Bluetooth 4.0 can reach up to 24 Mbps of transfer rate. At the same time, it has a low power consumption and is of small size. As such, the Bluetooth communication protocol is preferred to be embedded in portable devices than ZigBee. Bluetooth was therefore adopted as the wireless communication method in the proposed device.

The ECG data is transmitted to the Bluetooth module of the device through the SPI. The Bluetooth communication module obtains the ECG data package from the MCU and completes the Bluetooth data packet encapsulation of the ECG data. It presents the acquired ECG waveform through the mobile application (APP) via Bluetooth 4.0 in real-time.

## 3. Low-Power Design for the Core Devices

Low power consumption is one of the main requirements of wearable ECG equipment and is an important condition for ensuring long-term operation [29]. A wearable ECG device includes an ECG acquisition module, an embedded MCU, and a wireless communication module. The ECG acquisition module completes the amplification and performs the ADC processing of the ECG data. These steps are accomplished in a continuous working state. The embedded MCU is the control core of the entire device, while the wireless communication module wirelessly transmits the ECG data [30]. The power consumption of the entire wearable ECG device *P* is computed using Equation (1) below, while the entire circuit operation time *t* in hours can be calculated using the expression in Equation (2) below.
(1)$$P=P_{ADC}+P_{MCU}+P_{TRANS}+P_{0}$$
(2)$$t=\frac{C*V_{BAT}}{P}$$
where *P_ADC_*, *P_MCU_*, and *P_TRANS_* are the power consumption of the front-end ADC, MCU, and Bluetooth modules, respectively. *P_0_* is the power consumed by the power supply module, *C* is the capacity of the power supply battery, and *V_BAT_* is the battery voltage.

In this study, three ways were adopted to reduce the power consumption of the multi-lead wearable ECG device: Firstly, by optimizing the embedded MCU operation mode to reduce the MCU power, *P_MCU_*; secondly, by adjusting the radio frequency (RF) parameters in the process of wireless communication to reduce the wireless transmission power, *P_TRANS_*; thirdly, by using Huffman coding to compress the ECG data to reduce the amount of wireless communication data and the wireless transmission power, *P_TRANS_*. However, when compressing the ECG data, it increases the power consumption *P′* of the MCU. As such, the power consumption of the entire ECG device is modified using Equation (3), as follows:(3)$$P=P_{ADC}+P_{MCU}+\frac{1}{CR}P_{TRANS}+P_{0}+P^{\prime }$$
(4)$$\Delta P=P^{\prime }-(1-n)P_{TRANS}$$
(5)$$P^{\prime }<(1-n)P_{TRANS}$$
where *CR* is the compression ratio of the ECG data, *P′* is the increased power consumption due to the execution of the compression algorithm, and Δ*P* in Equation (4) is the difference between the power consumption before and after the data is compressed. Only by satisfying the expression in Equation (5) can the purpose of reducing power consumption be achieved using the Huffman compression algorithm. The optimization of the MCU operation mode, which allows it to operate in standby mode instead of active mode, is the first method to saving the operating power. When the MCU is in standby low-power mode, it can be woken up by the interrupt signal to execute the corresponding interrupt program. If the standby mode is exited during the interrupt routine, the MCU will operate in active mode when the interrupt routine is completed. It will then enter the standby low-power mode if it finds that there is no ECG data during the execution of the main program. The specific process is illustrated in Figure 6.

The wireless communication module is based on the Bluetooth 4.0 BLE (BT BLE) protocol stack for Bluetooth connectivity and data transfer. Therefore, the power consumption can be reduced by improving the parameters in the Bluetooth communication process.

First, two of the three default channels of the Bluetooth transmission are turned off and only broadcast on the 37 channels. The time interval between the transmissions of two adjacent broadcast packets is then set to 200 ms, which is twice the default value. The larger the broadcast interval, the smaller the power consumption. Moreover, in the connected state, the RF power consumption of the Bluetooth device is reduced, which requires an increase in the connection interval and the number of slave delays. However, that will slow down the rate of Bluetooth data transmission and lower the Bluetooth communication bandwidth. As a result, the experiment is carried out on the principle that the APP software will receive all the packets in order to find the most suitable slave delay and connection interval. Finally, the slave latency is set to 10, the connection interval is set to 20, and the monitoring timeout is set to 500.

As the number of ECG leads is increased, the amount of ECG data transmitted by the wearable ECG device via Bluetooth is also increased [31]. Compressing the ECG data can reduce the amount of wireless communication data, therefore reducing the power consumption of the wireless transmission, *P_TRANS_*. Among the data compression algorithms, this study chooses Huffman coding to realize the compression of the ECG data. Huffman coding is a lossless compression algorithm, which can restore the original data after decoding [32]. The Huffman coding considers the frequency and length of the code. Its principle is to represent high frequency data using a shorter code and low frequency data using a longer code.

The proposed ECG acquisition device has a sampling rate of 250 Hz, where each data sample is 3 bytes or 24 bits per lead. The ECG data format is stored as hexadecimal data, including the first byte (i.e., the high-eight-bit), the second byte (i.e., the middle-eight-bit), and the third byte (i.e., the low-eight-bit). The amount of three-lead data is 2250 bytes or 18,000 bits per second. The acquired ECG data from three leads are listed in Table 1. The data was hardly changed for the first byte in each lead. A slight change was contributed only when there was a huge baseline drift and motion artifact noise interference. For the second byte, the data was slightly changed, since the sampling rate of device was 250 Hz higher than the ECG frequency, which is low and is only within the range of 0.5 Hz to 3 Hz. Therefore, the adjacent two samples of data of the ECG signal also did not change that much. For instant, in lead I between the first sampling data of 32 and the second sampling data of 31, the data was just changed a little. For the third byte of these three leads, the data was changed more sensitively, since the ECG signal has a small variation. The ECG signal is a weak physiological signal; therefore, the change of data is mainly reflected in the low-level byte. The data of the first-byte and middle-byte has changes that were too small. As for the two high-level bytes, with 1500 bytes per second, data compression can be employed to reduce the amount of data.

The first method was to compress 1500 bytes of 1 s of ECG data by Huffman coding. In this case, when Huffman coding was performed, the amount of data processed and the Huffman binary tree established did not match the MCU memory. This consumed a long processing time. In order to solve this for the first compression method, the second method of 1500 bytes of ECG data was separately stored into 6 arrays. The 6 arrays were then separately compressed by Huffman coding. Table 2 shows the results of the compression using the second compression method. Each array had 250 bytes of original data, where the first byte of lead I was compressed to 10 bytes with a compression rate (CR) of 25. The results show that the CR of the original and compressed data for each lead is relatively high. As such, the second method of compression was adopted to compress the ECG data of the first and second byte of data for each lead. This method ensured a large CR that allowed a high sampling rate to be used for transmission via a Bluetooth communication module.

## 4. Experimental Results and Performance Analysis

The experimental results of the compression algorithm using the core equipment with Huffman code designed in this paper are presented. The participants were tested for a span of 60 s in three different states, including sitting, walking, and jogging. The data collected was stored in the mobile APP. Moreover, the data was imported into MATLAB simulator for decompression [33], verifying the correctness of the compression algorithm, as well as analyzing the effect in the compression under the three different states.

Figure 7a shows the ECG data in the sitting state after MATLAB. In addition to a slight baseline drift, the ECG waveform shows a clear waveform with less noise. This verifies that the compression algorithm can be implemented on the designed wearable ECG device proposed in this study. The ECG data in the walking state after MATLAB decompression is shown in Figure 7b. Motion artifact noise was introduced due to the walking motion. This is evident in lead I with smaller amplitudes. The ECG data in the jogging state after MATLAB decompression is shown in Figure 7c. It can be seen in the waveform that the noise is higher in the jogging state over the first two states. The features of lead I are submerged, and the P waves of lead II and V1 are also affected. By comparing and analyzing each state, it can be seen that in the sitting state, the CR is highest and the compression effect is best. However, irregularities in the waveform will result in a very low CR.

Figure 8a presents a participant sitting on a stool wearing the proposed ECG vest design. Through the ECG vest, the ECG signal was collected and sent to a smart phone. The real-time ECG waveform acquired through lead II was displayed via the mobile APP, as shown in Figure 8b. The ECG signal acquisition was tested again, this time using the patient monitor. The patient monitor device is a medical grade device produced by a medical device company. It can simultaneously measure the ECG signal of seven leads. The ECG of the three leads of I, II, and V1, measured by the patient monitor device and ECG vest designed in this work are shown in Figure 8c. It can be seen that the ECG waveforms of the three leads of I, II, and V1, collected by the proposed device, are clear and in par with the commercially-produced patient monitor device.

The operation current of the proposed device was tested under four conditions of a low power device design using 3.7 V lithium batteries as the energy supply. As listed in Table 3, the reduction in the operation currents under three different methods were 3.36 mA, 1.30mA, and 0.30mA. The operating power consumption of the entire device is reduced by 31% when all measures are performed.

Table 4 presents the performance comparison between the proposed design and previously proposed systems in the literature. In the case of the normal operation of the device, the output voltage measured by the power supply module is 2.10 V. According to the measurement result, the current consumption of the proposed low-power acquisition system is 9.87 mA (i.e., reduced by 37.6%), which can operate for 50 h using a 500-mAh battery. The sampling rate of the proposed system is 250 Hz per lead, which is suitable for clinical diagnosis, compared to other references [30,31,32]. Moreover, the power dissipation is smaller than in [27,34], which can achieve a long-term application.

The proposed device is a real-time three-lead ECG acquisition device with a Bluetooth wireless transmission system and is compared with a single-lead ECG acquisition device with ZigBee, as proposed in [10]. The proposed device in this study has several advantages as opposed to [10]. First, the transmission data of the proposed system is four times more than [10]. Second, the battery capacity is 500 mAh, which is 150 mAh less, as opposed to 650 mAh in [10]. Third, it uses Bluetooth communication protocol, which is more widely used than ZigBee for wearable devices used together with mobile phones. On the other hand, [10] using Zigbee has a current consumption and power dissipation of 2.4 times and 20.72 mW less than the proposed device in this study, respectively. Table 4 lists the performances of the previously proposed system and the proposed system in this study. The list of abbreviations used in this article is included in Table 5.

## 5. Conclusions

In this study, a multi-lead wearable ECG signal sensor system is designed for patients with cardiac diseases and users who need ECG monitoring. Firstly, the ECG vest is designed as an interface to connect the ECG device to the human body. Secondly, an analog front-end circuit, an MCU module, and a Bluetooth module are integrated into the core device. The entire wearable ECG device can finally complete a series of functions that involves collecting the ECG signals, processing and compressing the ECG data, and transmitting the ECG data via Bluetooth. A lossless compression scheme suitable for 2-byte ECG data was proposed with a sampling rate that is increased to 250 Hz per lead, similar to the clinical ECG holter. The proposed vest-type ECG system is more suitable in monitoring and nursing the off-ward patients. A series of tests were performed on the wearable ECG device. The experimental results show that the proposed device achieves a compression rate of 2.31 with an overall power consumption reduction of 37.6% compared with the pre-design. The design needs further improvement for irregular waveforms, which result in very low CR.

## Figures and Tables

**Figure 1 sensors-19-04996-f001:**
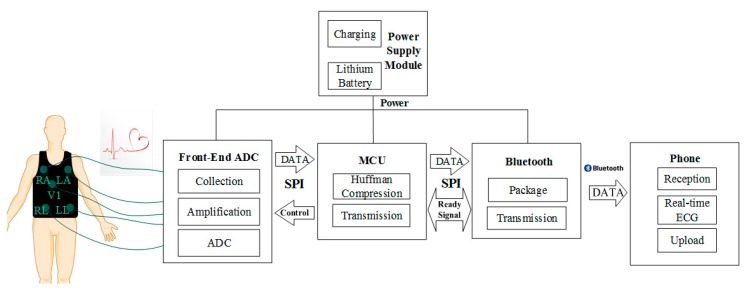
The structure diagram of the proposed multi-lead wearable electrocardiogram (ECG) sensor system.

**Figure 2 sensors-19-04996-f002:**
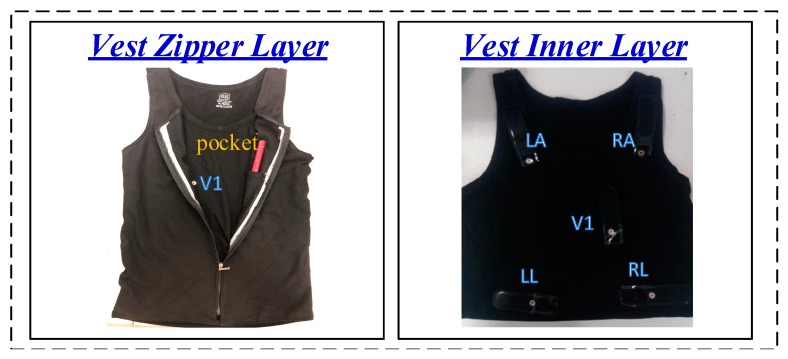
The overview of the ECG Vest.

**Figure 3 sensors-19-04996-f003:**
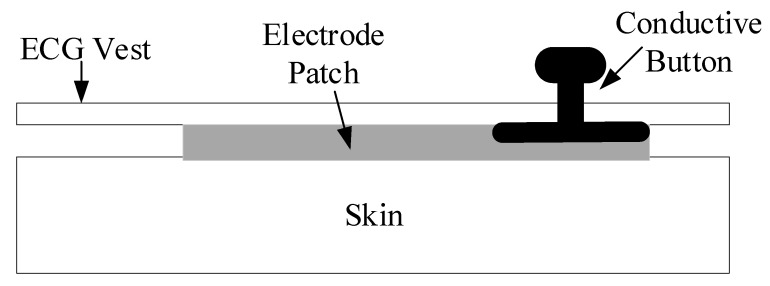
Illustration of the silica dry electrode design.

**Figure 4 sensors-19-04996-f004:**
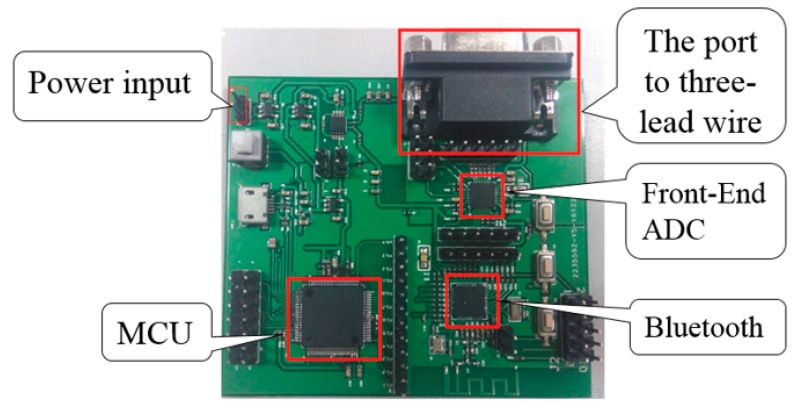
Hardware architecture implemented in the ECG vest.

**Figure 5 sensors-19-04996-f005:**
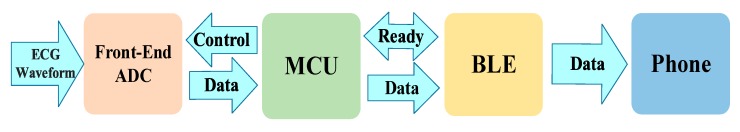
The connection diagram of the whole device.

**Figure 6 sensors-19-04996-f006:**
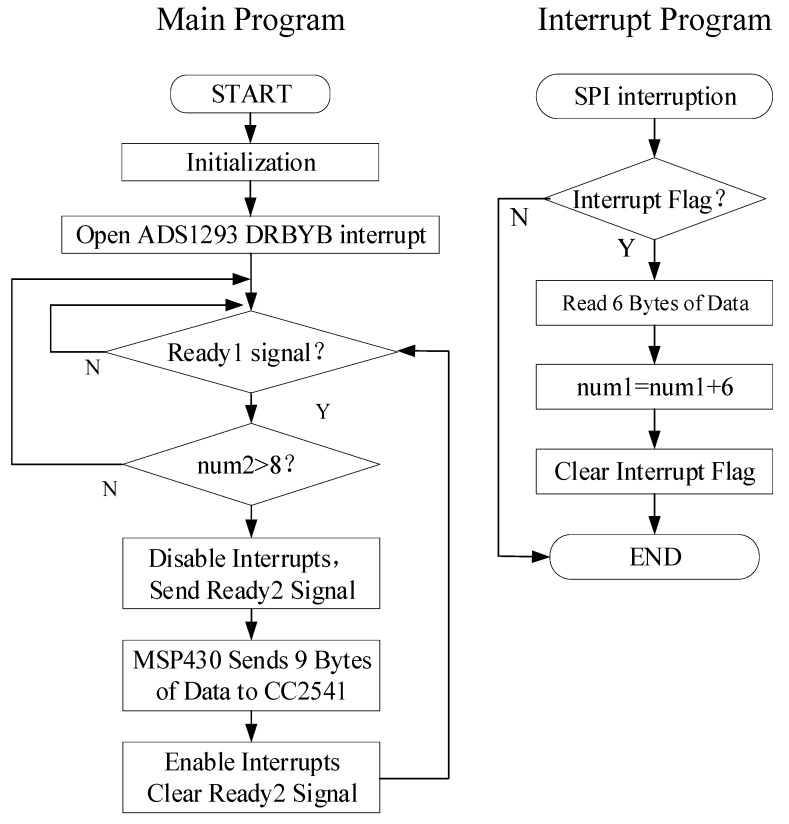
The overall flow chart of the micro control unit (MCU).

**Figure 7 sensors-19-04996-f007:**
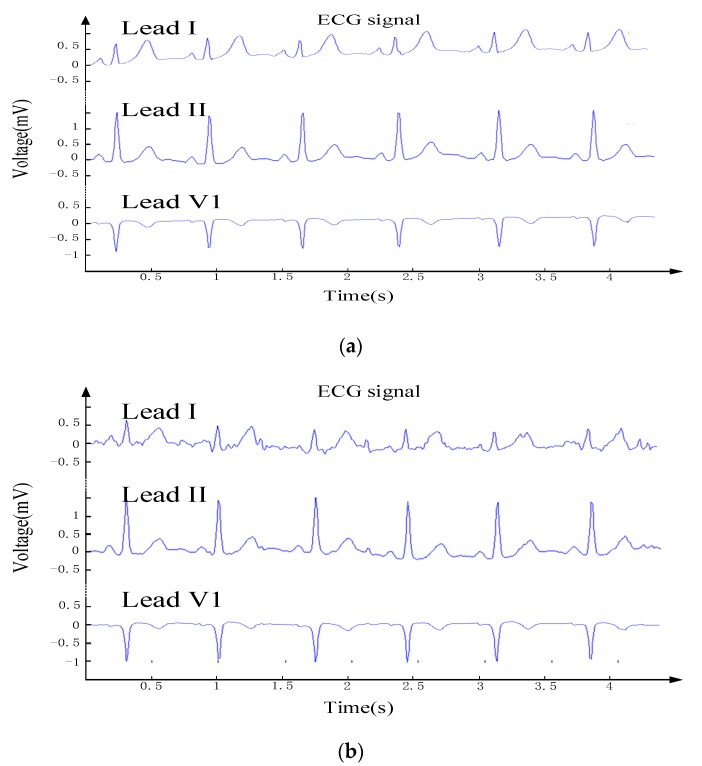
Sample ECG signal after decompression using MATLAB simulator; (**a**) in the sitting state; (**b**) in the walking state; and (**c**) in the jogging state.

**Figure 8 sensors-19-04996-f008:**
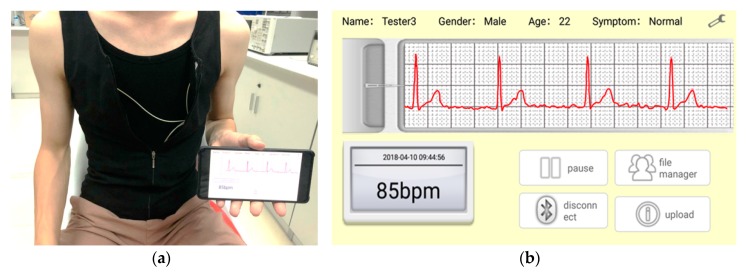
Illustrated is the following: (**a**) Usage of the ECG vest for ECG data acquisition and transmission to the mobile; (**b**) display of the acquired ECG signal on an Android-based smart phone; and (**c**) display of the acquired ECG signal on the monitor device of the patient.

**Table 1 sensors-19-04996-t001:** Part of the three-lead ECG data collected using the proposed device.

Byte Position per Lead	Sampling							
	1	2	3	4	5	6	7	8
The first byte of lead I	44	44	44	44	44	44	44	44
The second byte of lead I	32	31	31	31	31	31	31	31
The third byte of lead I	69	FE	95	6A	39	34	2F	0B
The first byte of lead II	42	42	42	42	42	42	42	43
The second byte of lead II	EC	EB	EB	EB	EB	EB	EB	EB
The third byte of lead II	02	CA	7A	67	6D	93	9D	A0
The first byte of lead V1	3F	3F	3F	3F	3F	3F	3F	3F
The second byte of lead V1	F2	F2	F2	F2	F3	F3	F3	F3
The third byte of lead V1	4C	65	82	F6	61	80	B8	D2

**Table 2 sensors-19-04996-t002:** Performances using the second compression form.

	Original Data (bytes)	Compressed (bytes)	CR
The first byte of lead I	250	10	25.000
The second byte of lead I	250	233	1.073
The first byte of lead II	250	48	5.210
The second byte of lead II	250	193	1.300
The first byte of lead V1	250	10	25.000
The second byte of lead V1	250	155	1.610
Total	1500	649	2.310

**Table 3 sensors-19-04996-t003:** Current (mA) of the low-power designed device using three different methods.

	Optimized	Non-Optimum	Current Reduction
MCU Operating Modes	12.45	15.81	3.36
Bluetooth Communication Parameters	14.51	15.81	1.30
Data Compression	15.51	15.81	0.30
Total	10.82	15.81	4.99

**Table 4 sensors-19-04996-t004:** Comparison of performances of previously proposed systems and the proposed system in this study.

		[34]	[27]	[10]	This Work
Power Dissipation (mW)	Front-End ADC	-	-	0.40	0.36
	MCU	-	-	-	1.11
	BT (BLE)	-	-	-	17.92
	Total	150.00	51.00	12.00	20.72
Current Consumption (mA)		41.80	17.00	4.07	9.87
Battery Capacity (mAh)		500	256	650	500
Life Time (h)		12	15	160	50
Sample Rate Per Lead (Hz)		100	500	320	250
ECG Leads		1	1	1	3
Reliability		Medium	-	High	High
Security		Medium	-	Medium	High
Popularity		Low	-	Low	High
Wireless Communication Protocol		ZigBee	Ant	ZigBee	BLE

**Table 5 sensors-19-04996-t005:** List of abbreviations used in this article.

Full Term	Abbreviation
Electrocardiogram	ECG
Micro Control Unit	MCU
Radio Frequency	RF
Internet of Things	IoT
Bluetooth	BT
Analog-to-Digital Converter	ADC
Bluetooth 4.0 Low Energy	BLE
Right Arm	RA
Left Arm	LA
Left Leg	LL
Right Leg	RL
Right-Leg Drive	RLD
Alternating Current	AC
Common Mode Rejection Ratio	CMRR
Micro Universal Serial Bus	Micro-USB
Serial Peripheral Interface	SPI
Central Processing Unit	CPU
Access Point	AP
Wireless Personal Area Network	WPAN
Application	APP
Compression Ratio	CR

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
