# Peer review of "A Low-Power High-Data-Transmission Multi-Lead ECG Acquisition Sensor System"

_sensors, 2019, doi:10.3390/s19224996_

Round 1

Reviewer 1 Report

General comment

Wang and colleagues describe the development of a novel multi-lead wearable ECG acquisition system, whose main feature is its low power design.

Major comments:

Abstract
The fields of application for the developed device should be clearly described in the abstract. Benefits and perspective associated with the developed system should be provided.

Introduction
Please provide further argumentation as for why multi-lead ECG has higher clinical value than single-lead ECG.

Section 3
Table 1 is particularly cryptic. The legend is meaningless. There are no headers and I found it hard to understand the content of the table.

Section 4
Figures 7, 8 and 9 could be grouped into one 3-panel Figure. Same applies to Figures 10, 11, 12.

Conclusion
Perspectives regarding the usefulness of the device in a more general context should be provided.

Minor comments

All abbreviations does not seem to be described. Please provide a table of abbreviations.

Several typos have been found e.g. spaces are missing at the end of the first sentence of the third paragraph of the Introduction or in the first upper left cell of Table 4. Please carefully check for syntax errors, text formatting and correct usage of English.

The table legends are too concise and some explanations of their content / meaning of abbreviations are not provided.

Author Response

Thank you very much for your review comments. Please find our response at the attachment.​

Reviewer 2 Report

The reviewer wishes to express his respect to the authors’ work.  But the reviewer thinks that readers will not catch the following points from the paper.

1.    Why do the authors choose the vest-type ECG acquisition system?  Many studies targeting such vest-type system and a wristband-type system are already published as is mentioned by the authors.

2.    The authors summarize the performance of the system in Table 4.  The reviewer doesn’t think that the system proposed by the authors is superior to that of Ref. [35] because the lifetime of battery is shorter than that of Ref. [35].  Although the total performance of the system must be evaluated based on its aim and purpose, this paper doesn’t mention the point.  Then readers will not think that the system proposed by the authors is superior to others.

3.    Although the authors mention “Although the power dissipation is larger than [35], the proposed wireless communication system is Bluetouth application which is widely used than ZigBee.”, this description is inappropriate because practical points are not addressed.

Author Response

(The authors gave the same response as above.)

Round 2

Reviewer 1 Report

The authors addressed my previous concerns in an appropriate manner.

Reviewer 2 Report

The reviewer approves that the authors have improved the paper following the comments of the reviewer.  The paper deserves publication.